# Emergence of Objectivity for Quantum Many-Body Systems

**DOI:** 10.3390/e24020277

**Published:** 2022-02-14

**Authors:** Harold Ollivier

**Affiliations:** Institut National de Recherche en Informatique et en Automatique, 2 Rue Simone Iff, 75012 Paris, France; harold.ollivier@inria.fr

**Keywords:** decoherence, quantum–classical transition, many-body system, quantum Darwinism

## Abstract

We examine the emergence of objectivity for quantum many-body systems in a setting without an environment to decohere the system’s state, but where observers can only access small fragments of the whole system. We extend the result of Reidel (2017) to the case where the system is in a mixed state, measurements are performed through POVMs, and imprints of the outcomes are imperfect. We introduce a new condition on states and measurements to recover full classicality for any number of observers. We further show that evolutions of quantum many-body systems can be expected to yield states that satisfy this condition whenever the corresponding measurement outcomes are redundant.

## 1. Introduction

The emergence of classical reality from within a quantum mechanical universe has always been central to discussions on the foundations of quantum theory. While decoherence—through interactions of a quantum system with its environment—accounts for the disappearance of superpositions of quantum states [1,2,3], it does not provide an a priori explanation for all intrinsic properties of a classical world and, in particular, for the emergence of an objective classical reality.

Quantum Darwinism [4,5,6,7,8,9,10,11] proposes a solution to fill this gap. Its credo states that rather than interacting directly with systems of interest, observers intercept a small fraction of their environment to gather information about them. Classicality then emerges naturally from quantum Darwinism. First, observing the system of interest S indirectly, by measuring its environment E rather than directly with an apparatus, restricts obtainable information to observables on S that are faithfully recorded in the environment. In practice, these observables are commuting with the well-defined preferred pointer basis induced by decoherence due to the interaction Hamiltonian between S and E. Second, requiring the observer to be able to infer the state of S by measuring only a small fraction of E implies that many such observers can do the same without modifying the state of the system. This, in turn, grants the state of the system an objective existence, as it can be discovered and agreed upon by many observers.

While early descriptions of quantum Darwinism [4,5] focused on simple models to build intuition, several subsequent works have studied the redundancy of information in more complex settings. References [8,12,13,14,15] show that quantum Darwinism—through the redundant proliferation of information about the pointer states in the environment—is a rather ubiquitous phenomenon encountered in many realistic situations.

The models used above to exemplify quantum Darwinism consider that the whole universe can be naturally split between S, the system of interest, and E, the environment itself subdivided into subsystems $E=\cup_{i}E_{i}$. As a consequence, the emergence of classicality is de facto analyzed relative to this separation. Redundant information is sought about observables on S in E. Yet, this is already going beyond what seems to be the minimal requirement that should allow to recover classical features of the universe: a natural egalitarian tensor–product structure for the state space, without explicit reference to a preferred system–environment dichotomy.

Such a scenario is particularly relevant for the Consistent Histories framework [16,17,18]. The universe is viewed as a closed quantum system in which one wants to identify a single set of consistent histories that describe the quasi-classical domain, where emergent coarse-grained observables follow the classical equations of motion [19], and become objective for observers embedded in the quantum universe. In a similar fashion, this scenario is adapted for understanding the emergence of objective properties in many-body physics. The reason is that for such composite systems, quantum fluctuations can be recorded into complex mesoscopic regions, e.g., in the course of their amplification by classically chaotic systems. Hence, redundant information need not be relative to observables of a single subsystem or any predefined set of subsystems, but rather to observables of to-be-determined sets of subsystems. Ref. [20] examines this question and shows that, due to the absence of a fixed set of subsystems defining the system of interest S, it is possible to construct redundant records for two mutually incompatible observables. While this gives a clear example where redundancy of information is not enough to guarantee the uniqueness of objective observables, the main result of [20] shows that this ambiguity requires the redundant records to delicately overlap with one another. In practical situations, such a delicate overlap is expected to be unlikely, thereby recovering the usual uniqueness of objective observables.

The present work shows that a similar conclusion can be expected in a more general setting, where redundant records are not required to be perfectly imprinted in the Hilbert space of the whole universe and where observables are replaced with POVMs (Throughout this paper, observable refer to sharp observables, so that POVMs are a generalization of observables). To this end, Section 2 presents an overview of [20] and outlines some of the key ingredients used implicitly when relying on perfect redundant records of observables. Section 3 generalizes the tools defining redundancy and classicality to our scenario. Section 4 provides a sufficient criterion on the approximate redundant records to recover classicality for a single set of POVMs on S. Finally, Section 5 takes a dynamical perspective to the emergence of objectivity and shows that our criterion is expected to hold in a wide range of situations, thereby implying that quantum Darwinism is a ubiquitous explanation for the emergence of classical properties in quantum many-body systems.

## 2. Objectivity for Idealized Quantum Many-Body Systems

In Ref. [20], an archetypal quantum many-body system is introduced to study the emergence of objective properties. It consists of a quantum system S composed of a collection of microscopic quantum systems $S=\cup_{i=1}^{N}S_{i}$. As a consequence, the Hilbert space $H_{S}$ of S has a natural tensor–product structure, $H_{S}={⨂}_{i}H_{S_{i}}$.

Objective classical properties for S are expected to emerge from redundant imprints that are accessible to observers using feasible measurements on fractions of S. More precisely, assuming S is in a pure state |ψ〉, redundant observables should induce a decomposition of |ψ〉 into orthogonal but un-normalized branches $|\psi_{i}\rangle$
(1)$$|\psi \rangle =\sum_{i}|\psi_{i}\rangle ,$$
each $|\psi_{i}\rangle$ being a common eigenstate of the redundant observables. This implies that, for measurements on fractions of S, this coherent superposition is indistinguishable from the incoherent classical mixture $\sum_{i}|\psi_{i}\rangle \langle \psi_{i}|$, thus forbidding observers to experience the quantumness of the correlations between fragments of S.

The similarity with quantum Darwinism should be clearly apparent: for both, not all subsystems can be measured simultaneously, thus forcing partial observations. In the presence of faithful redundant imprints, this would allow several observers to agree on their measurement results, thereby granting those records and associated observables an objective existence.

However, the similarity stops here. For quantum many-body systems, one cannot readily conclude that evolutions inducing faithful redundant imprints will favor the emergence of a *single* set of redundant observables, contrarily to usual system–environment settings [21]. The reason for such difference stems from the absence, in the many-body setting, of precise localization for the redundant records themselves.

For instance, in Ref. [21], although the choice of one subsystem of the whole universe for playing the role of reference system is arbitrary—any other would be equivalent for the purpose of the conducted analysis—it is clearly identified, and the redundant imprints refer to a measurement record of an observable for this specific subsystem. Therefore, comparisons between the conclusions drawn for different choices of the reference subsystem cannot be made. Even more strikingly, Ref. [20] gives a concrete example of two redundantly recorded, yet non-commuting, observables for S. One or the other could then equally pretend to be objective, while their combination does not allow the branch decomposition of Equation (Equation 1).

To see this, consider S made of qubits $S_{i,j}$ where (i,j)∈[1,N]×[1,N]. The state of S is prepared by applying a CPTP map Λ from a single qubit to S and defined in the following way:$$\begin{array}{cc} |0\rangle & \to |\bar{0}\rangle =\frac{1}{\sqrt{2^{N}}}\overset{N}{\underset{i=1}{⨂}}\left(\overset{N}{\underset{j=1}{⨂}}{|0\rangle }_{i,j}+\overset{N}{\underset{j=1}{⨂}}{|1\rangle }_{i,j}\right)\\ |1\rangle & \to |\bar{1}\rangle =\frac{1}{\sqrt{2^{N}}}\overset{N}{\underset{i=1}{⨂}}\left(\overset{N}{\underset{j=1}{⨂}}{|0\rangle }_{i,j}-\overset{N}{\underset{j=1}{⨂}}{|1\rangle }_{i,j}\right). \end{array}$$

Clearly, for fixed *i*, the measurement of the qubits labeled {(i,j),j∈[1,N]} in the basis ${({⨂}_{j=1}^{N}|0\rangle }_{i,j}\pm {⨂}_{j=1}^{N}{|1\rangle }_{i,j})/\sqrt{2}$ is equivalent to the measurement of the whole system relative to the basis $\{|\bar{0}\rangle ,|\bar{1}\rangle \}$. This means that the information about the observable $\bar{Z}=|\bar{0}\rangle \langle \bar{0}|-|\bar{0}\rangle \langle \bar{1}|$ is perfectly imprinted *N* times in S.

In addition, one can also rewrite the vectors $|\bar{0}\rangle$ and $|\bar{1}\rangle$: $$\begin{array}{cc} |\bar{0}\rangle \; & =\frac{1}{\sqrt{2^{N}}}\overset{N}{\underset{i=1}{⨂}}\left(\overset{N}{\underset{j=1}{⨂}}{|0\rangle }_{i,j}+\overset{N}{\underset{j=1}{⨂}}{|1\rangle }_{i,j}\right) \end{array}$$(2)$$\begin{array}{cc} & =\frac{1}{\sqrt{2^{N}}}\sum_{b=0}^{2^{N}-1}\overset{N}{\underset{j=1}{⨂}}|b_{j}\rangle \\ |\bar{1}\rangle \; & =\frac{1}{\sqrt{2^{N}}}\overset{N}{\underset{i=1}{⨂}}\left(\overset{N}{\underset{j=1}{⨂}}{|0\rangle }_{i,j}-\overset{N}{\underset{j=1}{⨂}}{|1\rangle }_{i,j}\right) \end{array}$$(3)$$\begin{array}{cc} & =\frac{1}{\sqrt{2^{N}}}\sum_{b=0}^{2^{N}-1}\overset{N}{\underset{j=1}{⨂}}{\left(-1\right)}^{b}|b_{j}\rangle , \end{array}$$
where, for a given *b* written as a binary string $b=(b_{1},\ldots ,b_{N})$, $|b\rangle ={⨂}_{i=1}^{N}{|b_{i}\rangle }_{i}$. Combining Equations (Equation 2) and (Equation 3), the conjugate basis has a simple expression:$$\begin{array}{cc} \frac{|\bar{0}\rangle +|\bar{1}\rangle }{\sqrt{2}} & =\frac{1}{\sqrt{2^{N}}}\sum_{\underset{h(b):\mathrm{even}}{b\in [0,2^{N}-1]}}\overset{N}{\underset{j=1}{⨂}}|b_{j}\rangle \\ \frac{|\bar{0}\rangle -|\bar{1}\rangle }{\sqrt{2}} & =\frac{1}{\sqrt{2^{N}}}\sum_{\underset{h(b):\mathrm{odd}}{b\in [0,2^{N}-1]}}\overset{N}{\underset{j=1}{⨂}}|b_{j}\rangle , \end{array}$$
where h(b) denotes the Hamming weight of *b*. As a consequence of this rewrite, for fixed *j*, any measurement of qubits labeled {(i,j),i∈[0,N]} that reveals the parity of the weight of $b_{j}$ is equivalent to a measurement of the conjugate observable $\bar{X}=|\bar{0}\rangle \langle \bar{1}|-|\bar{1}\rangle \langle \bar{0}|$. Hence, the information about $\bar{X}$ is perfectly imprinted *N* times in S, leading to an apparent paradox. Each $\bar{X}$ and $\bar{Z}$ defines a set of redundantly imprinted observables, yet each set is incompatible with the other. The measurement results that can be gathered by observers measuring the redundant imprints cannot be explained by resorting to a classical mixture of orthogonal states. Here, redundancy is not enough to imply the classicality of observables.

Nonetheless, it should be noted that both observables cannot be measured simultaneously by different observers in spite of their redundancy. This is because any redundant record of $\bar{X}$ and any reduncant record of $\bar{Z}$ overlap in exactly one qubit and require incompatible measurements for this specific qubit. Thus, it is not possible to share one redundant record of $\bar{Z}$ with one observer, and one about $\bar{X}$ with another. It is also not possible to have the first observer perform a non-destructive measurement of $\bar{Z}$ on its part of S and pass the overlapping qubit to the second observer so that he/she measures $\bar{X}$: the first measurement already destroys the needed coherence for the second.

This remark is the core of the main result of [20] for recovering objectivity for quantum many-body systems. A sufficient criterion is introduced to guarantee that any two redundant records in S, possibly corresponding to different observables *F* and *G*, can always be measured in any order and yet yield compatible results. More precisely, it ensures that the state |ψ〉 of the whole system S can be written as $|\psi \rangle =\sum_{i}|\psi_{i}\rangle$, where each $|\psi_{i}\rangle$ is a simultaneous eigenstate of *F* and *G*, thereby ensuring the orthogonality of the $|\psi_{i}\rangle$ and the indistinguishability between |ψ〉 and $\sum_{i}|\psi_{i}\rangle \langle \psi_{i}|$ for feasible measurements.

To make this formal (see [20] for details), suppose $F={\left\{F_{f}\right\}}_{f\in F}$ and $G={\left\{G_{g}\right\}}_{g\in G}$ are two sets of redundantly recorded observables on S with respect to the corresponding partitions F and G of the microscopic sites $S_{i}$ of S. This means that for each element f∈F, there exists an observable $F_{f}\in F$ on *f* that can be decomposed into projectors ${\left\{F_{f}^{\alpha }\right\}}_{\alpha }$ where α is an eigenvalue of $F_{f}$ such that
$$\forall \alpha ,\;\forall f^{\prime }\in F,\;F_{f}^{\alpha }|\psi \rangle =F_{f^{\prime }}^{\alpha }|\psi \rangle ,$$
and similarly for $G_{g}\in G$ on g∈G with projectors ${\left\{G_{g}^{\mu }\right\}}_{\mu }$ associated to eigenvalues μ of $G_{g}$. Then, a sufficient condition on F and G to ensure that results of $F_{f}$ on *f* are compatible with those of $G_{g}$ on *g*, for all values of *f* and *g*, is that for all $f,f^{\prime }\in F$, there exists g∈G, possibly depending on *f*, and $f^{\prime }$ such that $f\cap g=f^{\prime }\cap g=\emptyset$, and vice versa with the roles of F and G permuted. This property is called *non pair-covering* of F and G [20].

As a result, when F and G are not pair covering each other, we have
$$\begin{array}{c} \forall f,f^{\prime }\in F,\;\exists g\in G,\;\;\;F_{f}^{\alpha }G_{g}^{\mu }|\psi \rangle =F_{f^{\prime }}^{\alpha }G_{g}^{\mu }|\psi \rangle ,\\ \forall g,g^{\prime }\in G,\;\exists f\in F,\;\;\;G_{g}^{\mu }F_{f}^{\alpha }|\psi \rangle =G_{g^{\prime }}^{\mu }F_{f}^{\alpha }|\psi \rangle . \end{array}$$

In essence, this means that not only are there redundant imprints of the observables in F in the state |ψ〉 of S, but the redundancy remains even though $G_{g}\in G$ is actively measured or |ψ〉 is decohered as a result of tracing out g∈G (and the same with the roles of F and G permuted).

This is indeed enough to impose the commutation on the support of |ψ〉: using the same notation, for any *f* and *g*, the non-pair covering condition gives
$$\begin{array}{cc} \exists g^{\prime }, & f\cap g^{\prime }=\emptyset \\ \exists f^{\prime }, & f^{\prime }\cap g=f^{\prime }\cap g^{\prime }=\emptyset . \end{array}$$

Then,
(4)$$\begin{array}{cc} F_{f}^{\alpha }G_{g}^{\mu }|\psi \rangle & =F_{f}^{\alpha }G_{g^{\prime }}^{\mu }|\psi \rangle \end{array}$$
(5)$$\begin{array}{cc} & =F_{f^{\prime }}^{\alpha }G_{g^{\prime }}^{\mu }|\psi \rangle \end{array}$$
(6)$$\begin{array}{cc} & =G_{g^{\prime }}^{\mu }F_{f^{\prime }}^{\alpha }|\psi \rangle \end{array}$$
(7)$$\begin{array}{cc} & =G_{g}^{\mu }F_{f^{\prime }}^{\alpha }|\psi \rangle \end{array}$$
(8)$$\begin{array}{cc} & =G_{g}^{\mu }F_{f}^{\alpha }|\psi \rangle , \end{array}$$
where Equations (Equation 4) and (Equation 8) follow from the redundancy of records, Equations (Equation 5) and (Equation 7) derive from the non-pair covering condition, and Equation (Equation 6) is a direct consequence of the absence of overlap between $f^{\prime }$ and $g^{\prime }$.

One can now prove by induction that the same holds for multiple sets of redundantly imprinted observables F,G,…Z. Their projectors commute over |ψ〉, allowing to define a common branch decomposition for the state of the system as prescribed by Equation (Equation 1).

## 3. Approximate Records and Classicality for Quantum Many-Body Systems

The significance of the non-pair covering criterion introduced in the previous section is due to the relative ease with which it is met in practice. The overlap that is required to maintain the ambiguity between redundantly recorded, yet incompatible, observables is too delicate to happen in realistic physical systems—see [20] for an extended discussion on this point.

However, this reasoning suffers from several drawbacks. First, the non-pair covering criterion is applicable only to (sharp) observables and not to the broader information gathering strategies that can be implemented using POVMs. Second, redundant observables must be perfectly imprinted in fragments of S. Both restrictions can be ultimately traced back to how redundancy is measured and how classicality is deemed, that is, whenever projective measurements are compatible on the state |ψ〉 of the system, or equivalently, whenever they commute on the support of |ψ〉.

The paragraphs below address these two points by providing a definition of approximate redundant records of POVMs and an alternative witness for their classicality.

### 3.1. Approximate Copies of POVM Records

Let $S=\cup_{i=1}^{N}S_{i}$ be a many-body system with *N* microscopic sites. Denote by F a partition of [1,N] and by $S_{f}=\cup_{i\in f}S_{i}$, for f∈F.

**Definition** **1**(δ-approximate records). *For $f,f^{\prime }\in F$ with $f\neq f^{\prime }$, and two POVMs $F_{f}={\left\{F_{f}^{\alpha }\right\}}_{\alpha }$ and $F_{f^{\prime }}={\left\{F_{f^{\prime }}^{\alpha }\right\}}_{\alpha }$, respectively, on $S_{f}$ and $S_{f^{\prime }}$. For δ>0, we say that $F_{f^{\prime }}$δ-approximately records $F_{f}$ on the system state ρ if, ∀α,*
$$\mathrm{tr}\left(F_{f}^{\alpha }⊗F_{f^{\prime }}^{\alpha }\rho \right)\geq \left(1-\delta \right)\mathrm{tr}\left(F_{f}^{\alpha }\rho \right).$$

As expected, this definition captures the fact that, given that outcome α is observed by measuring $F_{f}$ on ρ, a measurement of $F_{f^{\prime }}$ yields the same outcome α with a probability of at least 1−δ. This is because
$$\mathrm{Pr}\left(F_{f^{\prime }}yieldsoutcome\;\alpha |F_{f}yieldsoutcome\;\alpha \right)=\frac{\mathrm{tr}(F_{f}^{\alpha }⊗F_{f^{\prime }}^{\alpha }\rho )}{\mathrm{tr}(F_{f}^{\alpha }\rho )}.$$

When the above property is true for all $f,f^{\prime }\in F$, we say that the set of POVMs $={\left\{F_{f}\right\}}_{f\in F}$ is *|F|-times δ-approximately redundant*.

The following lemma shows that Definition 1 falls back to that of [20] for δ=0, pure system states and (sharp) observables.

**Lemma** **1.**
*Assume $F_{f}$ and $F_{f^{\prime }}$ are projective measurements on disjoint subsets f and $f^{\prime }$ of [1,N], and that $F_{f^{\prime }}$ 0-approximately records $F_{f}$ on |ψ〉. Then*

$$F_{f}^{\alpha }⊗F_{f^{\prime }}^{\alpha }|\psi \rangle =F_{f}^{\alpha }|\psi \rangle .$$



**Proof.** Define the following normalized states
$$|\psi_{F_{f}^{\alpha }}\rangle =\frac{F_{f}^{\alpha }|\psi \rangle }{\mathrm{tr}(F_{f}^{\alpha }|\psi \rangle \langle \psi |)}and|\psi_{F_{f^{\prime }}^{\alpha }}\rangle =\frac{F_{f^{\prime }}^{\alpha }|\psi \rangle }{\mathrm{tr}(F_{f^{\prime }}^{\alpha }|\psi \rangle \langle \psi |)}.$$By assumption, $\mathrm{tr}(F_{f}^{\alpha }⊗F_{f^{\prime }}^{\alpha }|\psi \rangle \langle \psi |)=\mathrm{tr}(F_{f}^{\alpha }|\psi \rangle \langle \psi |)$. Using the definition of $|\psi_{F_{f}^{\alpha }}\rangle$, this becomes
$$\mathrm{tr}(F_{f^{\prime }}^{\alpha }|\psi_{F_{f}^{\alpha }}\rangle \langle \psi_{F_{f}^{\alpha }}|)\times \mathrm{tr}(F_{f}^{\alpha }|\psi \rangle \langle \psi |)=\mathrm{tr}(F_{f}^{\alpha }|\psi \rangle \langle \psi |).$$Hence, one concludes that $\mathrm{tr}(F_{f^{\prime }}^{\alpha }|\psi_{F_{f}^{\alpha }}\rangle )=1$, which implies that
(9)$$F_{f^{\prime }}^{\alpha }|\psi_{F_{f}^{\alpha }}\rangle =|\psi_{F_{f}^{\alpha }}\rangle .$$Similarly, for all α,β, we have
$$\mathrm{tr}(F_{f}^{\alpha }|\psi_{F_{f^{\prime }}^{\beta }}\rangle \langle \psi_{F_{f^{\prime }}^{\beta }}|)=\frac{\mathrm{tr}(F_{f}^{\alpha }⊗F_{f^{\prime }}^{\beta }|\psi \rangle \langle \psi |)}{\mathrm{tr}(F_{f^{\prime }}^{\beta }|\psi \rangle \langle \psi |)}.$$Using Equation (Equation 9) on the rhs above and recalling that $F_{f^{\prime }}^{\alpha }\times F_{f^{\prime }}^{\beta }=F_{f^{\prime }}^{\alpha }\times 1_{\alpha =\beta }$, we obtain
$$\mathrm{tr}(F_{f}^{\alpha }|\psi_{F_{f^{\prime }}^{\beta }}\rangle \langle \psi_{F_{f^{\prime }}^{\beta }}|)=\frac{\mathrm{tr}(F_{f}^{\alpha }|\psi \rangle \langle \psi |)}{\mathrm{tr}(F_{f^{\prime }}^{\beta }|\psi \rangle \langle \psi |)}\times 1_{\alpha =\beta }.$$For fixed β, taking the sum over α yields 1, because $|\psi_{F_{f^{\prime }}^{\beta }}\rangle$ is normalized and $\sum_{\alpha }F_{f}^{\alpha }=1$, so that we can conclude that $\mathrm{tr}(F_{f}^{\alpha }|\psi_{F_{f}^{\alpha }}\rangle )=1$. In turn, this implies that $F_{f}^{\alpha }|\psi_{F_{f^{\prime }}^{\alpha }}\rangle =|\psi_{F_{f^{\prime }}^{\alpha }}\rangle$ and we arrive at
$$F_{f}^{\alpha }|\psi \rangle =F_{f}^{\alpha }⊗F_{f^{\prime }}^{\alpha }|\psi \rangle =F_{f^{\prime }}^{\alpha }|\psi \rangle .$$□

### 3.2. Extending the Compatibility Criterion as a Witness for Classicality

As previously argued, one expects that quantum Darwinism for a many-body system S implies that (i) a preferred set of POVMs emerges from the sole requirement of being approximately redundantly recorded in the state of S, and (ii) these POVMs exhibit classicality.

The natural choice of witness for classicality is that observers accessing fragments of S will be able to explain all the correlations of their measurement results without the recourse to quantum correlations. In [20], this is required for arbitrary pure quantum states of the system, which translates into the ability of the preferred observables to induce a decomposition of the state |ψ〉 of S into a superposition of orthogonal branches $|\psi \rangle =\sum_{i}|\psi_{i}\rangle$, where each $|\psi_{i}\rangle$ is a common eigenstate of all observables in redundantly imprinted sets $O_{1},O_{2},\ldots$, i.e.,
$$\forall O\in O_{1}\cup O_{2}\cup \ldots ,\;O|\psi_{i}\rangle =\omega \left(i,O\right)|\psi_{i}\rangle ,$$
thereby defining the compatibility of all the observables of $O_{1}\cup O_{2}\cup \ldots$ on |ψ〉.

As anticipated, compatibility does not generalize straightforwardly to POVMs due to the absence of a meaningful equivalent to eigenstates of observables. Nonetheless, several options have been proposed in other contexts to understand and sometimes quantify the classicality of POVMs, namely through the introduction of commutativity, non-disturbance, joint-measurability and coexistence (see, for example, [22,23]). Our choice, justified below, for the substitute for compatibility is based on joint measurability.

**Definition** **2**(Joint-measurability). *Let O be a set of POVMs, and for O∈O denote its elements by ${\left\{O^{\omega }\right\}}_{\omega }$. The set O is jointly measurable if and only if there exists a POVM T with elements ${\left\{T^{\theta }\right\}}_{\theta }$ such that*
(10)$$\forall O\in O,\;\forall \omega ,\;O^{\omega }=\sum_{\theta }p\left(\omega |O,\theta \right)T^{\theta },$$
*where p(ω|O,θ) is a probability distribution for ω when O and θ are fixed.*

This definition states that all measurements in O can be simulated by first measuring *T* and then, depending on the obtained outcome θ and the chosen O∈O, by sampling ω according to the probability distribution p(ω|O,θ).

This choice is motivated by the operational approach promoted by quantum Darwinism. Observers can perform measurements, accumulate statistics and investigate correlations between them. When POVMs are jointly measurable, observers are able to interpret the correlations of measurement results through a simple marginalization process.

Joint measurability is further justified as a witness of classicality, as it rules out steering—a purely quantum phenomenon—(see [24] for a review). On the contrary, coexistence can reveal steering [25], and is therefore not an appropriate choice in our context. Additionally, non-disturbance suffers from drawbacks in light of quantum Darwinism: it is usually asymmetric, meaning that measurements need to be carried out in a precise order so as to not disturb one another. This ordering requirement contradicts our everyday experience of classical features obviously robust to the precise order in which measurements are performed. Finally, commutativity is shown to imply joint measurability [22], but the converse is in general not true. Hence, without further good reasons to rule out joint measurability, witnessing classicality through commutativity risks being too restrictive and, thus, potentially missing the emergence of objectivity.

Additionally, Proposition 1 of [22] shows that when restricted to projective measurements, joint measurability is indeed equivalent to the commutativity of observables. Thus, our choice of witness for classicality reduces to that of Ref. [20], as compatibility on the state of the system reduces to commutativity on its support.

Lastly, to obtain a useful criterion for classicality in our context, it needs to account for (i) approximations and (ii) systems whose evolutions practically restrict their attainable states to a subset of all possible density matrices. To this end, we note that the operator equality of Equation (Equation 10) is equivalent to a statement on probabilities of the outcomes computed for system states ρ that span the set of density matrices for S. This is because the trace function is an inner product for the real Hilbert space of Hermitian matrices. Hence, we can deal with (i) by stating that probability distributions are close to that obtained for jointly measurable POVMs, and (ii) can be accounted for by enforcing the relation only on the set D of attainable states.

**Definition** **3**(δ-approximate joint measurability over D). *Let D be a set of density matrices, δ≥0 and O a set of POVMs, where the elements of O∈O are ${\left\{O^{\omega }\right\}}_{\omega }$. The set O is δ approximately jointly measurable over D if there exists a POVM T with elements ${\left\{T^{\theta }\right\}}_{\theta }$ such that*
(11)$$\forall O\in O,\;\forall \omega ,\;\forall \rho \in D,\;\left|\mathrm{tr}\left(O^{\omega }\rho \right)-\sum_{\theta }p\left(\omega |O,\theta \right)\mathrm{tr}\left(T^{\theta }\rho \right)\right|\leq \delta ,$$
*where p(ω|O,θ) is a probability distribution for ω when O and θ are fixed.*

## 4. Recovering Joint Measurability

As seen in Section 2, redundancy is not enough to imply classicality. The absence of a natural, or preferred, way to group microscopic sites of a quantum many-body system allows information about incompatible observables to be redundantly recorded in the whole system. Although incompatible observables cannot be read off at the same time by multiple observers—so that this statement does not violate axioms of quantum mechanics—they can still collectively decide beforehand which one to recover.

In the case of perfect redundant records of projective measurements, the non pair-covering condition ensures that only a single set of compatible observables can be accessed by observers, thus corresponding to the everyday experience. Given our definitions of approximate records and the replacement of compatibility with approximate joint measurability, the question we have to address is whether non pair-covering is enough to guarantee the joint measurability of a single set of observables.

**Theorem** **1.**
*Let S be a quantum many-body system, such that there exists F, a partition of [1,N] of the microscopic sites $S_{i}$ of S. Let $F={\left\{F_{f}\right\}}_{f}$ be a set POVMs, where $F_{f}$ acts on f only and satisfies ∀α and $\forall f,f^{\prime }\in F$,*

$$\forall \rho \in D,\;\mathrm{tr}\left(F_{f}^{\alpha }⊗F_{f^{\prime }}^{\alpha }\rho \right)\geq \left(1-\delta \right)\mathrm{tr}\left(F_{f}^{\alpha }\rho \right),$$

*for some δ>0, and D a set of density matrices. Assume there exists G, a second partition, and $={\left\{G_{g}\right\}}_{g}$ with g∈G a second set of POVMs satisfying the corresponding approximate redundantly recorded condition stated above. Assume that F and G do not pair-cover each other, then for all f∈F and g∈G, $F_{f}$ and $G_{g}$ are δ-approximately jointly measurable on D.*


**Proof.** The non pair-covering condition imposes that
$$\begin{array}{cc} \; & \forall f,f^{\prime }\in F,\exists g\in G,s.t.f\cap g=\emptyset \;and\;f^{\prime }\cap g=\emptyset \\ \; & \forall g,g^{\prime }\in G,\exists f\in F,s.t.g\cap f=\emptyset \;and\;g^{\prime }\cap f=\emptyset . \end{array}$$For given f∈F and g∈G, using the non pair-covering condition, it is possible to choose $f^{\prime }\in F$ and $g^{\prime }\in G$ such that
$$\begin{array}{c} f\cap g^{\prime }=\emptyset =f^{\prime }\cap g^{\prime }\\ g\cap f^{\prime }=\emptyset =g^{\prime }\cap f^{\prime }. \end{array}$$Then, using redundancy and the disjointness conditions above, for all α, we obtain
$$\begin{array}{cc} \mathrm{tr}(F_{f}^{\alpha }\rho )\; & \geq \mathrm{tr}(F_{f}^{\alpha }⊗F_{f^{\prime }}^{\alpha }\rho )\\ \; & =\mathrm{tr}(F_{f}^{\alpha }⊗F_{f^{\prime }}^{\alpha }⊗\sum_{\nu }G_{g^{\prime }}^{\nu }\rho )\\ \; & \geq \left(1-\delta \right)\mathrm{tr}\left(F_{f^{\prime }}^{\alpha }⊗\sum_{\nu }G_{g^{\prime }}^{\nu }\rho \right), \end{array}$$
and similarly for all ν
$$\begin{array}{cc} \mathrm{tr}(G_{g}^{\nu }\rho )\; & \geq \mathrm{tr}(G_{g}^{\mu }⊗G_{g^{\prime }}^{\mu }\rho )\\ \; & =\mathrm{tr}(G_{g}^{\mu }⊗G_{g^{\prime }}^{\mu }⊗\sum_{\beta }F_{f^{\prime }}^{\beta }\rho )\\ \; & \geq \left(1-\delta \right)\mathrm{tr}\left(G_{g^{\prime }}^{\mu }⊗\sum_{\beta }F_{f^{\prime }}^{\beta }\rho \right). \end{array}$$We also have
$$\begin{array}{cc} \mathrm{tr}(F_{f}^{\alpha }\rho )\; & =1-\sum_{\beta \backslash \alpha }\mathrm{tr}\left(f_{f}^{\beta }\rho \right)\\ \; & \leq 1-\left(1-\delta \right)\sum_{\beta \backslash \alpha }\mathrm{tr}\left(F_{f^{\prime }}^{\beta }⊗\sum_{\nu }G_{g^{\prime }}^{\nu }\rho \right)\\ \; & =1-\left(1-\delta \right)(1-\mathrm{tr}\left(F_{f^{\prime }}^{\alpha }⊗\sum_{\nu }G_{g^{\prime }}^{\nu }\rho \right))\\ \; & =\left(1-\delta \right)\mathrm{tr}\left(F_{f^{\prime }}^{\alpha }⊗\sum_{\nu }G_{g^{\prime }}^{\nu }\rho \right)+\delta , \end{array}$$
and similarly for $\mathrm{tr}(G_{g}^{\nu }\rho )$.Combining both inequalities, we arrive at
$$\begin{array}{cc} \; & \forall \alpha ,\nu ,\;\left|\mathrm{tr}\left(F_{f}^{\alpha }\rho \right)-\mathrm{tr}\left(\sum_{\nu }F_{f^{\prime }}^{\alpha }⊗G_{g^{\prime }}^{\nu }\rho \right)\right|\leq \delta ,\mathrm{and}\\ \; & \forall \beta ,\mu ,\;\left|\mathrm{tr}\left(G_{f}^{\mu }\rho \right)-\mathrm{tr}\left(\sum_{\beta }F_{f^{\prime }}^{\beta }⊗G_{g^{\prime }}^{\mu }\rho \right)\right|\leq \delta . \end{array}$$This concludes the proof, as the probabilities of obtaining outcomes $F_{f}^{\alpha }$ and $G_{g}^{\mu }$ are δ-close to that obtained by measuring $F_{f^{\prime }}^{\alpha }⊗G_{g^{\prime }}^{\mu }$ followed by the appropriate post processing, consisting of summing over the outcomes of the ignored POVM. □

Hence, any pair of approximately redundantly recorded POVMs is approximately jointly measurable. The trouble to recover a perfect analogue to the ideal case with pure states and projective measurements is that pairwise joint measurability does not imply global joint measurability [26]. That is, for three POVMs, all pairs can be jointly measurable, but all three of them might not be the marginals of a single POVM. As a consequence, one cannot claim full classicality in such a situation.

Global joint measurability can nonetheless be obtained by strengthening the non pair-covering condition into non tuple-covering.

**Definition** **4**(non tuple-covering). *F,G,…,Z partitions of [1,N] are non tuple-covering each other iff, $\forall f\in F,\;g\in G,\ldots ,\;z\in Z,\;\exists f^{\prime }\in F,\;g^{\prime }\in G,\ldots ,\;z^{\prime }\in Z$ s.t.*
$$\begin{array}{cc} f^{\prime }\cap g & =f^{\prime }\cap g^{\prime }=\ldots =f^{\prime }\cap z=f^{\prime }\cap z^{\prime }=\emptyset \\ g^{\prime }\cap f & =g^{\prime }\cap f^{\prime }=\ldots =g^{\prime }\cap z=g^{\prime }\cap z^{\prime }=\emptyset \\ \; & \vdots \\ z^{\prime }\cap f & =z^{\prime }\cap f^{\prime }=z^{\prime }\cap g=z^{\prime }\cap g^{\prime }=\ldots =\emptyset . \end{array}$$

Using this definition, the following theorem allows to recover global joint measurability.

**Theorem** **2.**
*Let $F={\left\{F_{f}\right\}}_{f\in F},\;G={\left\{G_{g}\right\}}_{g\in G},\ldots \;Z={\left\{Z_{z}\right\}}_{z\in Z}$ be sets of δ-approximate redundantly recorded POVMs on the state of a quantum many-body system S, with F,G,…Z partitions of [1,N], the indices of the microscopic sites. If the partitions F,G,…Z do not tuple-cover each other, then for any f,g,…z, $F_{f},G_{g},\ldots Z_{z}$ are δ-approximately joint measurable.*


**Proof.** Given the non tuple-covering condition, one could appropriately replace any measurement of $F_{f},G_{g},\ldots Z_{z}$ by a measurement of $F_{f^{\prime }},G_{g^{\prime }},\ldots Z_{z^{\prime }}$. From there, the same proof technique as the one used for Theorem 1 applies. Using the said replacement of measurements, one arrives at a situation where all POVMs $F_{f^{\prime }},G_{g^{\prime }},\ldots ,Z_{z^{\prime }}$ act on different subsets of the microscopic sites. They are, thus, defining a global POVM with elements $F_{f^{\prime }}^{\alpha }⊗G_{g^{\prime }}^{\beta }⊗\ldots Z_{z^{\prime }}^{\zeta }$ from which the probabilities of the outcomes (α,β,…,ζ) can be δ-approximated through classical post-processing. This allows to conclude about the δ-approximate joint-measurability criterion for POVMs ${\left\{F_{f}\right\}}_{f\in F},{\left\{G_{g}\right\}}_{g\in G},\ldots ,{\left\{Z_{z}\right\}}_{z\in Z}$. □

## 5. Dynamical Approach to the Emergence of Classicality

The non pair-covering condition has an appealing property of being rather simple and allowing the recovery of objectivity for usual many-body physics experiments: pair-covering is too delicate to maintain for macroscopic systems containing possibly millions or billions of microscopic sites so that they would necessarily be exhibiting only usual classical properties.

On the contrary, the non tuple-covering seems a more complex, if not harder, condition to achieve. This, in turn, weakens considerably the above argument and, as a consequence, the reach of quantum Darwinism for quantum many-body systems. Yet, we prove below that this is not the case, and that quantum Darwinism is a ubiquitous mechanism to explain the emergence of a single set of approximately jointly-measurable POVMs.

The way to address this question is to take a dynamical view at the creation of the redundant imprints into the state of the quantum many-body system. More precisely, we need to acknowledge the fact that the redundant imprints—be they perfect or approximate—are the result of an evolution from some initial state of an initial uncorrelated system R. In other terms, it results from the transformation of a state σ∈D(R) to a state ρ∈D(S), where D(R) is the set of density matrices for R and similarly for S. The transformation can then be represented by a CPTP map Λ so that ρ=Λ(σ).

The structure of the correlations, and hence of the information, between R and S can be analyzed using the techniques pioneered in [21] and refined in [27]. Yet, these need to be recast to fit into the quantum many-body setting, as they have been developed in the system–environment context.

**Theorem** **3.***Let* Λ *be a CPTP map from D(R) to D(S), and wq,wf∈[1,N], with S=∪i=1NSi and wq+wf≤N. For all σ∈D(R), consider ϱ=Λ(σ) the state of a generic quantum many-body system that evolved from the initial preparation state σ through* Λ. *Then, there exists a subset q of [1,N] of size at most wq such that for all subsets f of [1,N]\q with size wf, and for all POVMs Ff={Ffα}α on f*
∀α,tr(Ffαϱ)−∑θp(α|Ff,θ)tr(Tqθϱ)≤δ,
*with δ=dR2ln(dR)wfwq, where Tq is a fixed POVM on q that does not depend on f nor σ, and where p(α|Ff,θ) is a classical probability distribution for α when Ff and θ are fixed that is independent of σ. Above, dR denotes the dimension of R.*

**Proof.** The proof will proceed in two steps. First, it will follow the steps of Theorem 2 of [27] to obtain a bound on the distance between the Choi-states of two specific channels, one being the channel Λ reduced to some sufficiently small subsets *f* and the other one being a measure and prepare channel from R to *f*. The second step will focus on the measurement done by the measure and prepare channel and show that it can be understood as a measurement on a subset *q* disjoint and independent of *f*.Consider a basis |i〉 of R and a fiducial reference system R′ isomorphic to R. Define the maximally mixed state |ψ〉 of RR′ as 1/dR∑i|ii〉RR′. The Choi-state of Λ is then ρ=(1R′⊗Λ(|ψ〉〈ψ|) (see, for example, [28]). We can now apply Proposition 1 of [27] to ρ. For wf,wq∈[1,N], there exists q⊆[1,N] of size wq and Ξq a quantum–classical channel on *q* such that
∀f⊆[1,N]\q,|f|=wf,maxΞf∈QCI(R′:f|q)Ξf⊗Ξq(ρ)≤S(R′)ρwfwq.Above, Ξf is a quantum–classical channel on *f* such that Ξf(X)=∑αtr(Ffα|α〉〈α|) for some POVM Ff on *f*; S(R′)ρ is the von Neumann entropy for the system R′ when the global state is ρ; and I(R′:f|q)Ξf⊗Ξq(ρ) is the quantum mutual information between R′ and *f* conditioned on *q* for the global state Ξf⊗Ξq(ρ)—note that to ease notation, the obvious identity operators will continue to be omitted. The interest of this proposition is that it constructs a subset *q* of microscopic subsystems of S of size at most wq such that, *irrespective* of the choice of another subset *f* of microscopic subsystems of size wf disjoint from *q*, the correlations between R′ and any observation on *f* through Ξf conditioned on an observation of *q* through Ξq can be made small. This means that observing *q* through Ξq extracts all there is to know about R′ so that it becomes uncorrelated with any further observation on *f*. By analogy with the classical case, Ref. [27] refers to the region *q* as a quantum Markov blanket. We can now use this bound to arrive at a statement of closeness between two Choi-states. More precisely, for Ξq implementing the POVM Tq={Tqθ}θ on *q* so that Ξq(X)=∑θtr(TqθX)|θ〉〈θ|, we have
(12)trf¯(Ξf⊗Ξq(ρ))=Ξf∑θpθρR′fθ|θ〉〈θ|,with
(13)pθ=tr(Tqθρ)
(14)ρR′fθ=1pθtrf¯q(Tqθρ),
where f¯ is the complement of *f* in [1,N]\q so that the system S decomposes into ff¯q. As a consequence, I(R′:f|q)Ξf⊗Ξq(ρ)=∑θI(R′:f)Ξf(ρR′fθ). Using the quantum Pinsker inequality [29] for I(R′:f)Ξf(ρR′fθ), one obtains that
12ln2Ξf(ρR′fθ−ρR′θ⊗ρfθ)12≤I(R′:f)Ξf(ρR′fθ).This being true for all θ, using the convexity of both the square function and the 1-norm, we obtain
12ln2Ξf(ρR′f−∑θpθρR′θ⊗ρfθ)12≤I(R′:f|q)Ξf⊗Ξq(ρ).Now, using Equation (Equation 5) and S(R′)≤log(dR), we have that for all quantum–classical channels Ξf on *f*:
(15)Ξf(ρR′f−∑θpθρR′θ⊗ρfθ)1≤2ln(dR)wfwq.Above ρR′f is the Choi-state corresponding to Λf obtained by reducing the channel Λ to *f*, while ∑θpθρR′θ⊗ρfθ defines Γf, corresponding to a measure and prepare channel from R to *f*, as its Choi-state is separable with respect to the R′f partition. Note that in Γf, the prepared states ρfθ are independent of the input of the channel.We can define two additional channels, ΛfΞf=Ξf∘Λf and ΓfΞf=Ξf∘Γf that correspond to the Choi-states Ξf(ρR′f) and Ξf(∑θpθρR′θ⊗ρfθ), respectively, and that are realized by measuring the output states of Λf and Γf with the POVM Ff={Ffα}α. Then, we have, for all Ξf∈QC
ΛfΞf−ΓfΞf⋄≤dRΞf(ρR′f−∑θpθρR′θ⊗ρfθ)1
which implies, as the diamond norm is the result of an optimization over all input states and because of Equation (Equation 15), that
(16)$$\forall \sigma \in D\left(R\right),\;{∥\Lambda_{f}^{\Xi_{f}}\left(\sigma \right)-\Gamma_{f}^{\Xi_{f}}\left(\sigma \right)∥}_{1}\leq d_{R}\sqrt{2\ln\left(d_{R}\right)\frac{w_{f}}{w_{q}}}.$$We almost arrive at our result and just need to give a more explicit interpretation to both states in the above equation. $\Lambda_{f}^{\Xi_{f}}\left(\sigma \right)=\sum_{\alpha }\mathrm{tr}\left(F_{f}^{\alpha }\Lambda \left(\sigma \right)\right)\left|\alpha \rangle \langle \alpha \right|$ is the state obtained after measuring Λ(σ) using $F_{f}$ acting on subset *f* of size $w_{f}$. To interpret the state $\Gamma_{f}^{\Xi_{f}}\left(\sigma \right)$, recall that the output of a given channel Φ from R to *f* can be inferred from its corresponding Choi-state $\rho_{R^{\prime }f}^{\Phi }$, using the simple identity $\Phi \left(\sigma \right)={\mathrm{tr}}_{R^{\prime }}\left(\rho_{R^{\prime }f}^{\Phi }\sigma^{T}\right)$. Therefore, we have
$$\begin{array}{cc} \Gamma_{f}^{\Xi_{f}}\left(\sigma \right)\; & ={\mathrm{tr}}_{R^{\prime }}\left(\sum_{\theta }p_{\theta }\rho_{R^{\prime }}^{\theta }\sigma^{T}\right)⊗\left(\sum_{\alpha }\mathrm{tr}\left(F_{f}^{\alpha }\rho_{f}^{\theta }\right)\left|\alpha \rangle \langle \alpha \right|\right)\\ \; & ={\mathrm{tr}}_{R^{\prime }}\left(\sum_{\theta }{\mathrm{tr}}_{f\bar{f}q}\left(T_{q}^{\theta }\rho \right)\sigma^{T}\right)⊗\left(\sum_{\alpha }\mathrm{tr}\left(F_{f}^{\alpha }\rho_{f}^{\theta }\right)\left|\alpha \rangle \langle \alpha \right|\right)\\ \; & =\mathrm{tr}\left(\sum_{\theta }T_{q}^{\theta }\Lambda \left(\sigma \right)\right)⊗\left(\sum_{\alpha }\mathrm{tr}\left(F_{f}^{\alpha }\rho_{f}^{\theta }\right)\left|\alpha \rangle \langle \alpha \right|\right), \end{array}$$
where we use Equation (Equation 12) to replace $p_{\theta }\rho_{R^{\prime }}^{\theta }$ with ${\mathrm{tr}}_{f\bar{f}q}T_{q}^{\theta }\Lambda \left(\rho \right)$. Note that for the states $\rho_{f}^{\theta }$ for varying θ are independent of σ so that $\mathrm{tr}(F_{f}^{\alpha }\rho_{f}^{\theta })$ can be rewritten as $p(\alpha |F_{f},\theta )$, a classical probability distribution for α, given $F_{f}$ and θ. Equation (Equation 16) can now be rewritten as
$$\forall \sigma \in D\left(R\right),\;\sum_{\alpha }{∥\mathrm{tr}\left(F_{f}^{\alpha }\Lambda \left(\sigma \right)\right)-\mathrm{tr}\left(\sum_{\theta }T_{q}^{\theta }\Lambda \left(\sigma \right)\right)\mathrm{tr}\left(F_{f}^{\alpha }\rho_{f}^{\theta }\right)∥}_{1}\leq d_{R}\sqrt{2\ln\left(d_{R}\right)\frac{w_{f}}{w_{q}}}.$$All derivations above are independent from the choice of subset *f* and of quantum-classical channel $\Xi_{f}$—or, equivalently, of $F_{f}$—as long as $w_{f}$ and $w_{q}$ are chosen such that $\delta =d_{R}\sqrt{2\ln\left(d_{R}\right)\frac{w_{f}}{w_{q}}}$ is small. This concludes the proof as
$$\forall \varrho \in D,\;\forall \alpha ,\;\left|\mathrm{tr}\left(F_{f}^{\alpha }\varrho \right)-\sum_{\theta }p\left(\alpha |F_{f},\theta \right)\mathrm{tr}\left(T_{q}^{\theta }\varrho \right)\right|\leq \sum_{\tilde{\alpha }}\left|\mathrm{tr}\left(F_{f}^{\tilde{\alpha }}\varrho \right)-\sum_{\theta }p\left(\tilde{\alpha }|F_{f},\theta \right)\mathrm{tr}\left(T_{q}^{\theta }\varrho \right)\right|\leq \delta .$$□

In effect, Proposition 1 of [27] identifies a fraction of S that contains all the information that can be accessed about the initial state σ after Λ has taken place. This then decoheres all other possible smaller fractions *f* of S disjoint from *q*. The consequence is that any measurement on such fractions can be implemented by first measuring *q* and then by post processing classically the result depending on the choice of measurement $F_{f}$ on *f*. This being true for all sufficiently small fractions *f* and any measurement on $F_{f}$, we recover the δ-approximate joint measurability for such measurements over the states that are dynamically created by Λ from any initial state σ. Hence, when observers are restricted to fractions *f*, quantum Darwinism yields objective properties of the system that can all be understood as stemming from a single classical measurement on the Markov blanket *q*.

## 6. Conclusions

The last section shows that generic evolutions of quantum many-body systems do systematically generate Markov blankets that capture all correlations between fragments of S. As a consequence, measurement results obtained by observers measuring fragments of S outside Markov blankets can be explained using classical correlations only. This implies that the non tuple-covering condition is generically satisfied for all partitions of S that contain the Markov blanket. Hence, while the non tuple-covering condition seemed an a priori more complex requirement to satisfy compared to the non pair-covering, as soon as Markov blankets are outside the reach of observers, quantum Darwinism can be invoked to recover robust classical objective properties of quantum many-body systems. This is a situation similar to that of system–environment settings, where Markov blankets are created generically by quantum evolutions and are responsible for objective classical reality [21,27]. Further analysis of the precise location and accessibility of Markov blankets in realistic settings is left for future work.

## Data Availability

Not applicable.

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
