# Peer review of "Emergence of Objectivity for Quantum Many-Body Systems"

_entropy, 2022, doi:10.3390/e24020277_

Round 1

Reviewer 1 Report

The manuscript "Emergence of objectivity for quantum many-body systems" elaborates a very interesting observation of [20], that it is possible for multiple different observables to be redundantly recorded in the environment, and a problem which observables can have this property, or, be "compatible". The manuscript generalizes concepts of [20], that was limited to projective measurements being perfectly imprinted into fragments of the environment.

In the manuscript, both these limitations are relaxed by the introduction of a notion of delta-approximate records (Def. 1) that can be defined for imperfect imprinting of POVMs. The "compatibility" is generalized here to a term called "joint-measurability" (Def. 2). The last generalization required replacing the criterion of non-pair covering of fragments for compatibility with a stronger requirement of non-tuple covering (Def. 4). These generalizations are accompanied by a result given in Corollary 1 from Theorem 3, stating, if I understand it correctly, that it is relatively "easy" for POVMs over sufficiently fine-grained fragmentation of the environment, to meet the approximate joint-measurability criteria.

The result of the manuscript is rather theoretical and touches very subtle aspects of which quantities can be jointly measured. The results broaden understanding of the very deep properties of physical reality and deserve publication in the Special Issue of Entropy.

Yet, some parts of the manuscript are difficult to follow, and I recommend complementing them. I mean here theorem 3 and its discussion. The proof is very difficult to follow. I recommend rewriting it, in a more step-by-step way, especially regarding the part that applies the theorem of [27]. Some parts of the notation used in the proof are also not clear. More discussion of this result will also be in order.

Minor remarks:
- Lack of punctuation in the equation between lines 99 and 100.
- Misconstruction "that: such that" in line before equation before line 135.
- Definition 1 begins with sentence equivalent.
- Why, in Definition 1, do F_f and F_f' sets have different indexes (alpha and beta), if they refer to the same observable, just read from different fragments?
- In Definition 3 there should be "over D" before "iff".
- Some linking word is missing after referring to [26] in 239.

Author Response

I thank the referee for his careful reading and useful
  feedback. Following hist comment regarding the proof and discussion of
  theorem 3, this section has been completely rewritten.

  The idea behind the previous version was to acknowledge that a large
  part of theorem 3 was a consequence of theorem 2 of QiRanard21. Yet,
  as was obvious to the referree, using theorem 2 directly was difficult
  as I needed to pull some additional details that were obtained while
  prooving the theorem itself -- namely the measurement on the Markov
  blanket that captures all the possible correlations between their
  reference subsystem $A$ and small fractions $R$ of the environmental
  subsystems $\{B_i\}_i$.

  In this version, I resorted to only apply Proposition 1 of QiRanard21
  that gives the explicit construction of the Markov blanket. The rest
  still follows the same line of arguments as the ones used in their
  proof, but it now had the possibility to be tailored to the examined
  scenario.  In particular, I now incorporate corrolary 3 of the
  previously submitted version into the theorem itself. As consequence,
  theorem 3 of the revised draft now directly gives the
  $\delta$-approximate joint-measurability for observables on fractions
  of size $w_f$ outside the Markov blanket.                                                                                                 

  Overall, the proof should be much clearer, but also more informative
  as it better highlights the process responsible of the emergence of
  classicality: Proposition 1 of QiRanard21 constructs the Markov
  blanket. Applying it to the Choi-state of the channel together with
  Pinsker inequality allows to bound the distance between two channels,
  one being measure and prepare. This measure and prepare channel is
  then analyzed and shown to result from the evolution of the whole
  system followed by a fixed POVM on the Markov blanket. This
  measurement can then be identified as the one needed to recover any
  measurement result on small fractions of subsystems outside the Markov
  blanket followed by classical post-processing. That is we recover
  joint-measurability.

  I believe this new version was a necessary improvement over the
  initial text, and thank again the referee for asking for it.

  Minor remarks:
  - All typos have been dealt with.
  - Regarding Definition 1 and the different indices for the two
    observables $F_f$ and $F_{f'}$: there was no specific reason to
    distinguish the two sets of indices except than to make it clear
    that the elements of $F_f$ and $F_{f'}$ are not necessarily labelled
    in a coherent way initially -- i.e. they could use letters from
    different alphabets. But the definition of approximate records
    itself allows to relabel them in a coherent way so that both are
    within the same alphabet. Overall, I think this is a distraction
    from the main idea of what these approximate records are, and
keeping different letters for indices is indeed confusing. It is
    then removed from the current version.

  Additional remark:
  - Following the report of referee 2, part of the notation has been
    changed. See the reply to referee 2 for details.

Reviewer 2 Report

This manuscript presents a criterion for (approximate) joint measurability of POVMs in a many-body system. As a non-expert in quantum darwinism I understand this to be an important pre-requisite for the emergence of quasi-thermalized or classical states in a closed system. By doing so, the author generalizes the argumentation of Ref. [20] (which appropriately is paraphrased in the beginning of the manuscript allowing for an easier access to the content) to a more general scenario and shows how the results found there are recovered when assuming projective measurements. The problems arising from considering the more complicated case of POVMs are discussed and solved again in an easy to digest way.

I find the results shown in this manuscript important, nicely presented and therefore recommend publication as it is with minor points to be considered listed below.  

Definition 1: F -> F_{f’} = …, it would be nice to use Omega for POVM as down below to distinguish from projectors (or the other way around, in the current version it jumps back and forth)

Page 7 first line: yields

Paragraph before 174 (and following occasions): why the double indexing in the set \{\omega_j\}_j ?

Formula (11): missing a ‘for all j’?

Theorem 1: again F for POVM. I think it is not necessary to distinguish between F and G (allowing to use \Omega) as they always come with indices f and g. But this is only subjective preference.

Formula after line 233: G_g^\mu (or preferably \Omega_g^\mu)

Theorem 3: d_R undefined, from context I suppose it is the dimension of R?

Author Response

I thanks the referee for his appreciative comments and for the
  pinpointing of improvements to the readbility of the paper.

  Apart from the rewrite of Theorem 3 (see details in the reply to
  referee 1), the remarks on:
  - Definition 1
  - Paragraph 174,
  - Theorem 1
  have been taken care of by changing the notation to have a more
  appearent distinction between the various objects used in the paper.

  More precisely:
  - POVMs are now latin capital letters,
  - sets of POVMs are bold latin capital letters
  - sets of microscopic sites are lowercase latin letters
  - partitions of microsites are curly latin capital letters
  - channels are greek capital  letters
  - POVM elements are indexed with lowercase greek letters.

  Regarding the paragraph 174 (now 173), indeed we do need a double
  indexing as we have sets of POVMs each corresponding to the same
  information that is being redundantly recorded in the system. Within
  each set, we have an additional index that is relative to the element
  of the partition of $\mathcal S$ where the information is
  recorded. This double indexing being cumbersome, I tried to limit it
  as much as possible. I used the sets F, G, ... Z to avoid adding this
  index of sets in other instance. However, I still feel this is fine
  here, but I could very well replace it with an "explicit" list of sets
  F, G, ... Z if the referee feels it is more adequate. Additionally, I
  tried to emphasize the distinction between the sets (using the bold
  font and sometime also mentionning the word set) and the observables
  to avoid possible confusion elsewhere.

  All other comments have been implemented.

Round 2

Reviewer 1 Report

The revised version of the manuscript "Emergence of objectivity for quantum many-body systems" has been significantly improved in readability and, therefore, I recommend publishing it in the special issue of Entropy journal.

Here are my minor remarks that should be applied to the text:
- In the equation above line 129 there probably should be "for all alpha and f'" instead of "for all f and f'".
- There is a missing subscript "f'" in the second POVM in Definition 1.
- In line 214 there should be "O" instead of "j".
- Typos "ssteamming" in line 303.

Regarding Theorem 3 the space should be rewritten explicitly as $S=\cup_{i=1}^{N} S_i$ to remind the meaning of $N$, and confirm that it is used in this role. If there is a relation between $w_q$ and $w_f$ (e.g. $w_q + w_f \leq N$), then it should be stated. It would be consistent with the rest of the text if a subscript $q$ is used for POVM $T$, viz. $T_q$. The meaning of $d_\mathcal{R}$ should be provided in the body of the theorem, not in the proof.

Author Response

Dear Referee,

I've implemented all suggested changes pointed in your comments. Regarding the relation between $w_q$ and $w_f$, there is no additional constraint except that $q$ and $f$ are disjoint which implies $w_q + w_f \leq N$. Of course, depending on the other parameters (in particular $d_{\mathcal R}$) some values will give a very loose bound.

Best.